# Genetic Diversity and Population Structure of *Rhodeus uyekii* in the Republic of Korea Revealed by Microsatellite Markers from Whole Genome Assembly

**DOI:** 10.3390/ijms25126689

**Published:** 2024-06-18

**Authors:** Kang-Rae Kim, So Young Park, Ju Hui Jeong, Yujin Hwang, Heesoo Kim, Mu-Sung Sung, Jeong-Nam Yu

**Affiliations:** 1Animal & Plant Research Department, Nakdonggang National Institute of Biological Resources, Sangju 37242, Republic of Korea; kimkangrae9586@gmail.com (K.-R.K.); cindysory@nnibr.re.kr (S.Y.P.); juhii22@nnibr.re.kr (J.H.J.); dbwls8@nnibr.re.kr (Y.H.); heesookim@nnibr.re.kr (H.K.); 2Muldeuli Research, Icheon 12607, Republic of Korea; rkrtlqndj@naver.com

**Keywords:** genome assembly, *Rhodeus uyekii*, population genetics, bottleneck, microsatellite

## Abstract

This study is the first report to characterize the *Rhodus uyekii* genome and study the development of microsatellite markers and their markers applied to the genetic structure of the wild population. Genome assembly was based on PacBio HiFi and Illumina HiSeq paired-end sequencing, resulting in a draft genome assembly of *R. uyekii*. The draft genome was assembled into 2652 contigs. The integrity assessment of the assemblies indicates that the quality of the draft assemblies is high, with 3259 complete BUSCOs (97.2%) in the database of Verbrata. A total of 31,166 predicted protein-coding genes were annotated in the protein database. The phylogenetic tree showed that *R. uyekii* is a close but distinct relative of *Onychostoma macrolepis*. Among the 10 fish genomes, there were significant gene family expansions (8–2387) and contractions (16–2886). The average number of alleles amplified by the 21 polymorphic markers ranged from 6 to 23, and the average PIC value was 0.753, which will be useful for evolutionary and genetic analysis. Using population genetic analysis, we analyzed genetic diversity and the genetic structures of 120 individuals from 6 populations. The average number of alleles per population ranged from 7.6 to 9.9, observed heterozygosity ranged from 0.496 to 0.642, and expected heterozygosity ranged from 0.587 to 0.783. Discriminant analysis of principal components According to the analysis method, the population was divided into three populations (BS vs. DC vs. GG, GC, MS, DC). In conclusion, our study provides a useful resource for comparative genomics, phylogeny, and future population studies of *R. uyekii*.

## 1. Introduction

Bitterling fish are classified into two genera and 14 species, including nine endemic species [1]. *Rhodeus uyekii*, belonging to the subfamily Acheilognathinae, is distributed throughout most of South Korea [2]. The Honamkwon region is a large region that includes Chungcheongnam-do, Jeollanam-do, and Jeonbuk-do [3].

Korean rose bitterling *R. uyekii* is endemic in the Republic of Korea and a common freshwater fish that inhabits rivers and tributaries such as Geumgang, Mangyeong, Yeongsan, Seomjin, and Tamjingang rivers in the Honamkwon region [1]. This region accommodates genetic and species differentiation, including significant genetic differentiation between the Geumgang and Mankyungang Rivers in *Pseudopungtungia nigra*, new speciation of *Acheilognathus somjinensis* between genera Acheilognathus, the discovery of a new species of *Kichulchoia brevifasciata*, *Liobagrus geumgangensis*, and the genetic structure of the species in the genus *Coreoleuciscus* [4,5,6,7,8]. These regions are considered high-diversity hotspot areas. Therefore, uncovering the genetic structure of the species distributed in the Honamkwon region is important in evolutionary and genetic terms. In particular, given the significant genetic differentiation between the Geumgang and Mangyeonggang River populations of *Pseudopungtungia nigra*, it is expected that genetic differentiation has also occurred in populations between the Mangyeonggang and Geumgang River species of *R. uyekii* [4].

Understanding the genetic structure of a population has important implications for the use and conservation of organisms [9]. Additionally, being attractive is required for development as a variety. This species was proposed as a candidate for ornamental fish development due to its small size and beautiful body color [10]. Because *R. uyekii* has high value for its ornamental potential, it is necessary to identify the population structure to make it into a more valuable cultivar [10,11]. High genetic diversity of the population is advantageous for breed improvement [12]. Differences such as body color variation and genetic health between populations are helpful in breed improvement [12]. In the case of fighting fish, wild specimens are neither beautiful nor attractive [12]. In the case of *R. uyekii*, the wild individual is beautiful, but it is likely to be even more attractive if it is a cultivar. Therefore, basic research for breed improvement must be supported [12]. Here, basic research includes the whole genome sequence (WGS) of the species, mitochondrial DNA, microsatellite DNA to measure genetic diversity, and single nucleotide polymorphism [13,14,15]. The study of WGS is one of the most basic studies [13]. Such research is essential in many research fields, including genes, markers, and genetics [13,14,15,16]. Therefore, among these basic researches, research on WGS is essential [13]. For breed improvement, GWAS and genomic selection are used, but for these studies, the entire genome must first be studied [13,14,15]. Additionally, microsatellite markers are needed to measure the genetic diversity of wild populations and select populations to be improved into breeds [11].

Basic research on *R. uyekii* includes mitogenome sequencing and development studies on microsatellite markers [16,17]. However, the whole genome sequence, which is important information for breed improvement, has not been studied or reported, and no population genetic study has been conducted on *R. uyekii*.

Therefore, in this study, we plan to assemble the whole genome sequence to create a draft assembly, characterize and reveal the structure of the genome, and conduct research on whether there is genetic differentiation in population genetics for breed improvement. 

## 2. Results

### 2.1. Genome Assembly of R. uyekii

For whole-genome sequencing, 979,241,650 reads were generated using Illumina short-read sequencing, and the total read length was 147,865,488,978 bp. After removing adapter sequences and low-quality reads using Trimmomatic, the total number of clean reads was 504,366,512, and the total length of reads was 76,000,788,304 bp (Table 1). The Q20 and Q30 quality were 99.50% and 97.77%, respectively (Table 1). PacBio sequencing identified 5,573,756 HiFi reads, with a total length of 43,361,492,973 bp and an average HiFi read length of 3842 bp (Table 1). Genome assembly yielded 2652 contigs with a total length of 894,559,954 bp (Table 2). The N50 value was 2,964,661 bp. 

The K-mer distribution is shown in Figure 1 as a histogram constructed using Jellyfish software (ver. 2.3.0). The estimated haploid genome size is approximately 0.636 Gb, and the actual assembled genome size is 0.895 Gb (Table 3). 

Complete BUSCO analysis at BUSCO was confirmed (97.2%). BUSCO (Vertebrata odb10) analysis identified a total of 3259 BUSCO. Of these, 96.0% (3220) of the Vertebrata gene sequences were complete (Table 4), which comprised 3220 single-copy BUSCO (96.0%) and 39 duplicated-copy BUSCOs (1.2%). The number of fragmented BUSCOs was 35 (1.0%), and the number of missing BUSCOs was 60 (1.8%). 

### 2.2. Annotation of Candidate Genes and Proteins Prediction

For protein-coding genes, MAKER was used, and 31,166 were predicted. The total length of these genes was 85,997,067 bp (Table 5). Gene annotation across various databases resulted in the identification of 29,896 (95.93%), 24,820 (79.64%), 23,354 (74.93%), 9704 (31.14%), and 17,614 (56.52%) in EggNOG, InterPro, Pfam, COG, and KEGG databases.

In EggNOG, 29,896 genes were classified into 25 categories, excluding the extracellular structures category (Figure 2). Among these, the unknown function was the most common in 15,810 cases (51.93%). This was followed by intracellular trafficking, secretion, and vesicular transport in 2776 (9.12%), and posttranslational modification, protein turnover, chaperones in 2210 (7.26%), signal transduction mechanisms in 1569 (5.15%), transcription in 1505 (4.94%), replication, recombination and repair in 1132 (3.72%). 

In COG, 9704 genes were classified into 24 categories, excluding the extracellular structures and nuclear structure categories (Figure 3). Among these, signal transduction mechanisms in 1446 cases (14.90%). This was followed by General function prediction only in 1176 (12.12%), and cell cycle control, cell division, chromosome partitioning in 1064 (10.96%), translation, ribosomal structure and biogenesis in 969 (9.99%), posttranslational modification, protein turnover, chaperones in 832 (8.57%), Replication, recombination and repair in 426 (4.39%). 

In KEGG orthologs, 17,614 genes were annotated into 18 categories (Figure 4). Most of the genes annotated to “Genetic Information Processing” (32.49%), followed by “Environmental Information Processing” (15.45%), “Signaling and Cellular Processing” (13.24%). 

### 2.3. Phylogenetic Inference Orthologous Groups of R. uyekii 

We identified 31,396 orthogroups matching 602,402 genes Using Orthofinder, including *R. uyekii* and 10 species (Appendix A). The phylogenetic tree showed that *R. uyekii* and *Onychostoma macrolepis* belonged to the same clade, but distantly related. We analyzed the expansion and contraction of gene families based on the data of Orthogroups generated by Orthofinder (Figure 5). Expansion and contraction were analyzed based on statistically significant values of orthologroups. There were significant gene family expansions (8–2387) and contractions (16–2886) among the 10 fish genomes. *R. uyekii* showed more contraction (2886) than expansion (211). 

### 2.4. Genetic Diversity and Bottleneck Test

In microsatellite DNA, repetitive regions were found in 2005 contigs out of 2652 contigs (Appendix A). As a result of randomly selecting 100 primers, amplifying and genotyping, 21 microsatellite markers were screened (Appendix A). Twenty-one microsatellite markers were deposited in GenBank (OR753515-OR753535).

The 21 microsatellite loci were analyzed for genetic diversity indices in the six populations (Table 6). The number of average alleles ranged from 7.6 to 9.9, the *H*_O_ ranged from 0.496 to 0.642, and the expected heterozygosity ranged from 0.587 to 0.783. We find that all populations deviate from HWE. The inbreeding index was observed to be above 0.107 in all populations, the *F*_IS_ was significant (*p* < 0.05). The observed heterozygosity was highest in the GG population (*H*_O_ = 0.642) and the lowest in the DG population (*H*_O_ =0.496), but the observed heterozygosity of all populations did show differences.

Using the infinite mutation model (IAM), we identified significant bottlenecks in the All populations (*p* < 0.05). The TPM model was identified as a bottleneck in the BS and MS populations (Table 7). 

### 2.5. Population Structure and Genetic Differentiation Analyses

Most of the *F*_ST_ values in the microsatellite dataset were significant, with the highest *F*_ST_ value between DG and GG (*F*_ST_ = 0.193). The GC population had lower *F*_ST_ values in GC and DC (*F*_ST_ = 0.041). *R. uyekii* mostly showed *F*_ST_ values, indicating medium genetic differentiation (Table 8). Bayesian clustering analysis maximized delta *K* values for population structure at *K* = 2 (Figure 6). At *K* = 2, the populations were divided into two genetic populations. Unlike the STRUCTURE analysis results, the discriminant analysis of principal components (DAPC) results, which can analyze the structure of the population based on a nonmodel, show that the population (BS vs. DC vs. GG, GC, MS, DC) is divided into three populations (Figure 7).

## 3. Discussion

### 3.1. The Whole Genome Sequence of R. uyekii Provides a Useful Genetic Resource

If the first discovery of a species is considered the first milestone, then the whole genome sequence could be a landmark for applications in multiple fields [18]. Advances in next-generation sequencing (NGS) have made genomics research significantly higher throughput, cheaper, and more accessible [18,19]. Therefore, genome-level research has become more accessible, and research reports on it have increased [20,21,22]. The whole genome sequence is basically your genetic heritage [23]. This heritage is very important as a basis for applying genome-wide research, microsatellite marker development, and population genetic studies [4,11,23].

Fish account for more than half of the world’s vertebrate species [23]. The Cyprinidae family is the most diverse freshwater fish family in the world, with at least 210 genera and over 2010 species [24]. Until 2018, published genomic data were available for only 60 fish species [25]. Except for fish of economic value, there are currently no WGS projects underway on fish. WGS has been assembled primarily from economically important fish such as Atlantic salmon [26], carp [27], and channel catfish [28].

In this study, we reported for the first time the whole genome sequence of *R. uyekii*, a species of endemic species and ornamental value. The genome assembly of this fish was assessed as being of good quality (Table 2 and Table 4) and is expected to be a useful resource for further research on this fish of ornamental value and to provide baseline data for molecular breeding.

We combined the genomic dataset of *R. uyekii* obtained here with the other 10 genome datasets of Cyprinidae downloaded from NCBI to reconstruct the phylogenetic relationships within Cyprinidae at the genome level (Appendix A). The Acheilognathinae subfamily group has not been systematically studied, and no studies have yet investigated the monophyly or phylogenetic relationships of the Acheilognathinae subfamily. Therefore, the taxonomic position of the family Cyprinidae was reconstructed at the genome level. *Rhodeus uyekii* was found to be relatively closest to *Onychostoma macrolepis*, but due to the branching of the phylogenetic tree, it appears to be a fairly distant relative.

### 3.2. Genetic Diversity and Population Structure of R. uyekii

Genetic diversity, which is the evolutionary potential and ability to cope with environmental changes, has important implications [9]. In this study, we measured the genetic diversity of 120 individuals from six populations using 21 microsatellite markers of *R. uyekii*. The measured genetic diversity was *H*_O_ = 0.496–0.642, showing a moderate level of genetic diversity compared to other carp families [29,30]. Acheilognathinae fishes exhibit extensive genetic diversity [29,30]. Because *Rhodeus uyekii* has lower genetic diversity (*H*_O_ = 0.766) than *Acheilognathus koreensis*, it may have problems with its ability to adapt to environmental changes and its evolutionary potential. To solve these problems, methods for establishing conservation strategies for genetic diversity should be considered.

Inbreeding depression has been linked to population survival and reproduction [9,31]. Inbreeding increases homozygosity and deleterious allele frequencies, ultimately leading to a loss of genetic diversity [32]. In all populations, HWE escaped and the inbreeding coefficient (*F*_IS_) was significant (*p* < 0.001), indicating high inbreeding. This suggests that inbreeding occurred in all populations. The method of expected heterozygosity overestimation in IAM has recently been suitable for estimating bottlenecks. IAM results showed that all populations experienced a recent bottleneck (*p* < 0.05). Additionally, the BS and MS populations showed significant values using TPM (*p* < 0.05). This thus supports the recent emergence of bottlenecks in all populations. Declining population sizes and bottlenecks are caused by human activities such as habitat destruction and pollution [8]. Previous bottlenecks may have been caused by simple sampling bias due to human activities and small sample sizes [33]. Further studies with larger population sizes are needed. Nonetheless, the six populations involved in this study appear to be experiencing a bottleneck. Therefore, conservation efforts are needed to establish legal conservation areas to prevent habitat destruction due to human activities. In addition to these conservation efforts, a system for stably breeding populations in out-of-habitat breeding farms for breed improvement is needed. These measures could serve as a reservoir of genetic resources for *R. uyekii* when it is threatened with extinction in the wild, and this reservoir would reduce the risk of extinction [9]. It also suggests that such measures are necessary because, through breeding in captivity, complete extinction of species in the wild can be prevented.

The genetic structure of the six populations for *R. uyekii* was found to be *K* = 2 in STRUCTURE. However, since the nonmodel DAPC method is more accurate than STRUCTURE, we showed that DAPC divides into three populations. This is because significant genetic differentiation between Geumganggang and Mangyeonggang River was reported in *Pseudopungtungia nigra*, and similar results were found in *R. uyekii* with genetic differentiation between Geumganggang and Mangyeonggang River [4].

*Coreoleuciscus splendidus*, similar to *R. uyekii*, inhabits most water bodies in Republic of Korea. *Coreoleuciscus splendidus* was reported to be divided into two populations in the Honamkwon region basin [7]. In this study, the *R. uyekii* population showed significant genetic differences, and the genetic structure was divided into two populations (DG, BS vs. GC, GG, MS, DC) in the STRUCTURE results. However, DAPC results showed that the population was divided into three (BG vs. DG vs. GC, GG, MS, DC) on a nonmodel basis. Freshwater fish of the Korean Peninsula typically exhibit genetic variation across geographic watersheds, with little genetic variation within the same watershed [7]. The *F*_ST_ results showed significantly higher genetic differentiation in DG vs. BS, *F*_ST_ = 0.177, suggesting that the six populations are genetically three populations. This means that genetic differentiation exists between Geumgang and Mangyeonggang rivers, which is strongly supported in *Pseudopungtungia nigra* [4]. Therefore, it is necessary to divide the wild populations of *R. uyekii* into three conservation management in the Honamkwon area water system. Conservation of a species’ genetic structure is important in breed improvement [34]. This value is because genetically different groups may have differences in body color or unique patterns. Therefore, the implications of conserving populations in the wild suggest that they are of great value for breed improvement.

## 4. Materials and Methods

### 4.1. Sample Preparation

Samples were collected from wild populations located at (Gaesan, GS; 38°22′01.00″ N, 128°30′32.2″ E). Samples of pectoral fin for NGS analysis were collected from the Gaesan population. The collected samples were placed in 99.9% ethanol and stored in a deep freezer (−80 °C). Eight individuals from three populations were sampled for microsatellite marker polymorphism analysis. Genomic DNA was extracted using a DNeasy Blood & Tissue Kit (QIAGEN, Hilden, Germany) according to the manufacturer’s instructions. The quality of the extracted genomic DNA was checked using a 1% agarose gel and a Nanodrop spectrophotometer. Total RNA was extracted from leaf tissue using the RNeasy Mini Kit (QIAGEN) according to the manufacturer’s protocol (Macrogen, Inc., Seoul, Republic of Korea).

A total of 120 individuals were collected from six populations for use in population genetic analysis (Appendix A, Figure 8). The collected samples were placed in ethanol 99.9% and stored in a deep freezer (−80 °C). The quality of the extracted genomic DNA was checked using a 1% agarose gel and a Nanodrop spectrophotometer (NanoDrop ND1000, NanoDrop, Wilmington, DE, USA).

### 4.2. Genome Sequencing and de Novo Assembly

We used single-molecule real-time sequencing from Pacific BioSciences (PacBio, Menlo Park, CA, USA) HiFi following a protocol from Macrogen (Macrogen Inc., Seoul, Republic of Korea). The HiFi SMRTcell library was prepared using the SMRTBell Express Template Prep Kit 2.0 (PacBio, CA, USA). gDNA was cut into 6–20 kb fragments by g-TUBE (Covaris, Woburn, MA, USA) and reagents included in the Template Prep Kit were used. It also repaired DNA damage and fragment ends. A SMRTbell hairpin adapter was attached to the repaired extremity. AMPure PB beads (PacBio, CA, USA) were used for library enrichment and purification. SMRTbell template sizes greater than 10 kb were selected using the BluePippin system (SageScience, Beverly, MA, USA). HiFi reads generated from PacBio SequelIIe system were assembled using Genome assembly HGAP4 of single-molecule real-time (SMRT) link v. 11.0.0.146107 (NovogeneAIT, Singapore). Since this method already includes error correction of long reads, we continued polishing it with Illumina short reads, which have lower base calling errors. For Genome assembly, all the options used in analysis were set as default. In order to solve the reliability of the long reads determined base sequence, Illumina short reads sequencing was performed. The whole genome was sequenced using a 150 bp paired-end library produced by Macrogen (Macrogen Inc., Seoul, Republic of Korea) using the Illumina HiSeq 2500 (Illumina, San Diego, CA, USA) platform. The Illumina short read is the Trimmomatic ver. 0.33 [35] Clean reads were obtained after removing adapter sequences. To improve the accuracy of initial contig assembly, BWA-MEM [36] and Pilon version 1.23 [37] were used. Draft genomes generated in two assemblers were performed for the integrity of assembly with vertebrata odb version 10 using Benchmarking Universal Single-Copy Orthologs (BUSCO) ver. 5.2.2 [38]. The assembled genome has been deposited in NCBI GenBank whole-genome WGS database under accession number JAWRVZ000000000.

RNA Iso-seq libraries were prepared for genome annotation and sequenced using the PacBio sequel II platform according to the manufacturer’s protocol (Macrogen Inc., Seoul, Republic of Korea). For gene prediction, Isoform RNA sequencing (Iso-Seq) data were generated from PacBio Sequel II system. To generate high-quality transcripts, cDNA primer and a spurious false positive signal was carried out using lima v. 2.6.99. PolyA tail trimming and concatermer removal was performed to generate full-length non-concatemer reads using Isoseq3 v. 3.8.1 (https://github.com/PacificBiosciences/IsoSeq, accessed on 5 August 2023). After that, Clustering was performed to make the polished isoforms using isoseq3 cluster. 

Then, three tools, GlimmerHMM v. 3.0.4 [39], AUGUSTUS v. 3.3.2 [40] and SNAP v. 2.31.83 were used to make gene training models with Cypriniformes proteins downloaded from NCBI on June 2023 and high-quality transcripts data. After that, Maker v. 2.31.83 [41] was used to predict the gene model of the sample using the gene training model resulting from SNAP. 

### 4.3. Genome Annotations and Gene Annotation Analysis

Putative protein-coding genes in the genome sequence were predicted using the MAKER pipeline [41], using predicted Iso-Seq data of *R. uyekii* and peptide sequences from the genomes of *Carassius auratus*, *Cyprinus carpio*, *Rhinichthys klamathensis*. Two training sets, AUGUSTUS [40] and SNAP, were also used for prediction, and sequences with annotation edit distance score less than 0.5 were selected as high-confidence genes. Further annotation of consensus sequences against the InterPro (ver. 69.0), Pfam (ver. 31.0) and EggNOG (ver. 4.5) databases was performed using BLAST (ver. 2.7.1+, BLAST e-value: 1.0 × 10^−5^). For functional classification of annotations, Clusters of Orthologous Groups (COG) annotations were performed. KEGG (Kyoto Encyclopedia of Genes and Genomes) annotations were used for KEGG pathway mapping using KAAS [42].

### 4.4. Phylogenetic Tree Reconstruction

Genomes compared a total of 10 protein sequences, including *R. uyekii*, to other fish species. Protein sequences were downloaded from NCBI and used for analysis (Appendix A). Protein sequences were obtained from OrthoFinder ver. 2.5.4 [43] to organize groups of orthologous genes. After selecting groups from the phylogenetic tree using CAFE5 ver. 5.0.0 [44], Orthogroup expansion and contraction were measured using CafePlotter (https://github.com/moshi4/CafePlotter, accessed on 20 August 2023) reconstructed.

### 4.5. Analysis of Microsatellite Markers Genotyping

Microsatellite screening was performed using the MIcroSAtellite (MISA) tool (https://webblast.ipk-gatersleben.de/misa/, accessed on 23 August 2023). Parameters for di-, tri-, tetra-, penta- and hexa-nucleotides were set at 10, 4, 4, 4 and over 4 repeats. For each microsatellite loci, primers were designed according to the following five parameters using Primer3 software (https://github.com/primer3-org/primer3, accessed on 23 August 2023). The amplicon length was 100–400 bp, primer size was 20–24 bp, GC content was 40–60%, and the melting temperature was 58 °C. For the designed microsatellite primers, the presence of multiple binding regions in contigs was confirmed using SnapGene (GSL Biotech, Chicago, IL, USA).

Microsatellite loci amplification was performed using the Mastercycler® Pro Gene Amplifier (Eppendorf, Hamburg, Germany). The polymerase chain reaction condition was performed with a total volume of 20 μL using H-Star Taq DNA Polymerase (Biofact, Daejeon, Republic of Korea). PCR fluorescent labeling was performed according to the protocol described by Schuelke [45] using four fluorescent primers: microsatellite-specific forward primer (0.4 μM), reverse microsatellite synthesized with an M13 (TGTAAAACGACGGCCAGT) tail at the 5′ end, Specific primer (0.8 μM), fluorescent label (6-FAM, VIC, NED, PET) M13 primer (0.4 μM). The PCR conditions were 94 °C for 5 min, 94 °C for 30 s, 56 °C for 45 s, 72 °C for 45 s, 94 °C for 30 s, 53 °C for 45 s, 30 times, 12 times at 72 °C for 45 s, and 72 °C for 10 s. The final extension was for 10 min. For microsatellite PCR amplification products, GeneScan™ 500 ROX Size Standard Ladder (Applied Biosystems, Foster City, California, CA, USA) was mixed with HiDi™ formamide, denatured at 95 °C for 2 min and incubated at 4 °C. Allele sizes were determined using an ABI 3730xl DNA Analyzer (Applied Biosystems). Genotyping was determined using the GeneMarker® 2.6.7 program (SoftGenetics, State College, PA, USA). To evaluate the usefulness of the developed microsatellite loci, polymorphism information content (PIC), number of alleles, predicted heterozygotes (*H*_E_) and observed heterozygotes (*H*_O_) were analyzed using Cervus 3.0 software [46].

### 4.6. Genetic Diversity and Population Structure Analysis

The MICROCHECKER software v. 2.2.3 [47] was used to examine the presence or absence of scoring errors in the microsatellite loci. Genetic diversity was measured as the number of alleles (*NA*), expected heterozygosity (*H*_E_), and observed heterozygosity (*H*_O_) using the CERVUS software v. 3.0 [46]. The population inbreeding coefficient (*F*_IS_) and Hardy-Weinberg equilibrium (HWE) variance analyses were performed using GENEPOP v. 4.2 [48] and ARLEQUIN software v. 3.5 [49]. Two methods were used to estimate bottlenecks. The first method involved the BOTTLENECK software v. 1.2.02 [50], a program for estimating bottlenecks through heterozygous excess testing, and the infinite allele model (IAM) [51]. A two-phase model (TPM) and stepwise mutation model (SMM) [52] were used to estimate, and TPM was performed with 10% variance and 90% SMM. In addition, each model had 10,000 iterations, and significance was verified using the Wilcoxon signed-rank test [52]. 

The ARLEQUIN software v. 3.5 [49] was used to analyze the differences in genetics between groups as well as analyze molecular variance (AMOVA). The STRUCTURE software v. 2.3 [53] was used to perform genetic structure clustering analysis based on the Bayesian method model. To estimate the most suitable population, we set the population constant (*K*) to 1–10, and a suitable admixture model was applied to the mixture of water systems. The burn-in period was repeated 10 times with 10,000 iterations, and Markov chain Monte Carlo with 100,000 iterations. To estimate a population-appropriate constant (*K*), we analyzed a study by [53] and the cluster results corresponding to *K* using STRUCTURE HARVESTER [54]. A discriminant analysis of principal components (DAPC) was performed on the population using the R package ADEGENET v. 2.1.3 [55], a nonmodel-based genetic clustering method.

## 5. Conclusions

This is the first report to characterize the genome of *R. uyekii* and analyze the genetic structure of *R. uyekii* in water systems in the Honamkwon region basin of Republic of Korea to establish conservation strategies. Six populations of *R. uyekii* showed significant genetic differentiation in the microsatellite dataset. The habitat of *R. uyekii*, which lives in the water system of the Honamkwon region water system, is divided into three populations, so it must be divided into three management units due to its genetic structure. This study is the first report to study the development of characteristic microsatellite markers and the application of markers from *R. uyekii* genome information. Assembly of the genome generated a draft genome assembly of *R. uyekii* based on PacBio Sequel II and Illumina paired-end sequencing. The draft genome was assembled from 2652 contigs with an estimated genome size of 0.895 Gb. The integrity evaluation of the assembly revealed 3259 complete BUSCOs (97.2%) in Vertebrata odb10, indicating the high quality of the draft assembly. A total of 31,166 protein-coding genes were successfully predicted and annotated in the protein database. The phylogenetic tree showed that *R. uyekii* is a close but distant relative of *Onychostoma macrolepis*. Among the ten fish genomes, there were significant gene family expansions (8–2387) and contractions (16–2886). *R. uyekii* had greater contraction (2886) than expansion (211). The average number of alleles amplified by the 21 polymorphic markers was 6–23, with an average PIC value of 0.753, which we believe will be useful for evolutionary and genetics analyses. The number of alleles ranged from 7.6 to 9.9, the *H*_O_ ranged from 0.496 to 0.642, and the expected heterozygosity ranged from 0.587 to 0.783. The discriminant analysis of principal components (DAPC) results, which can analyze the structure of the population based on a nonmodel, show that the population (BS vs. DC vs. GG, GC, MS, DC) is divided into three populations. We conclude that these markers can strongly support genetic diversity analysis and cultivar development studies as basic data. In conclusion, our study provides a useful resource for comparative genomics, phylogeny, and future population studies of *R. uyekii*.

## Figures and Tables

**Figure 1 ijms-25-06689-f001:**
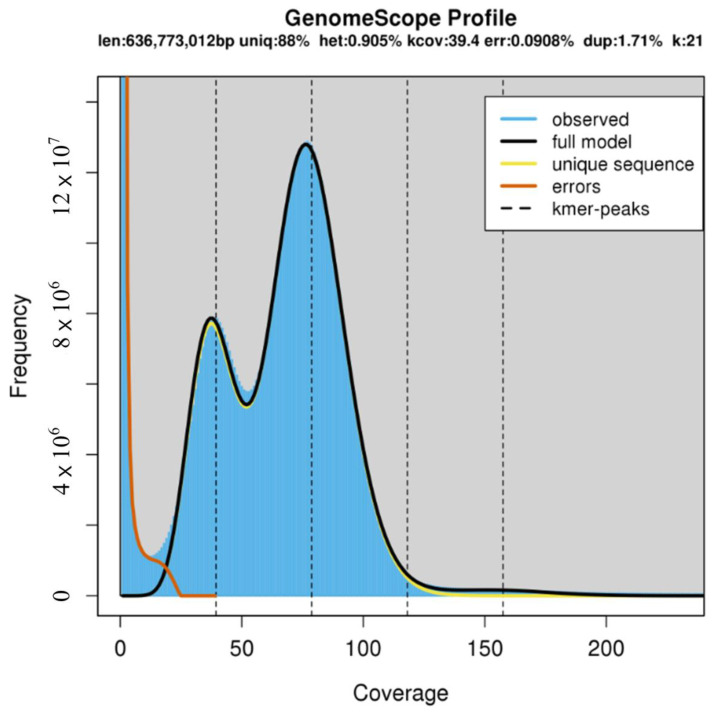
Estimation of the genome size of *R. uyekii*.

**Figure 2 ijms-25-06689-f002:**
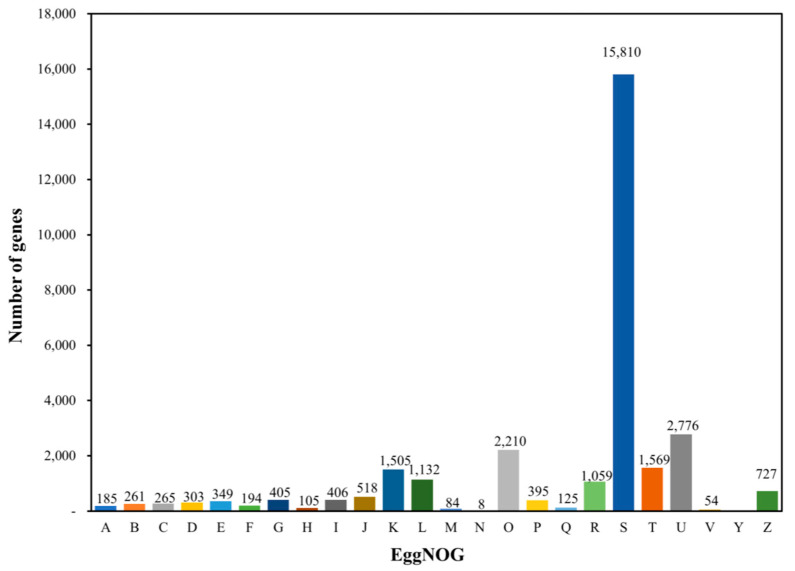
EggNOG functional classification information in *R. uyekii*. Genes were assigned to 25 categories.

**Figure 3 ijms-25-06689-f003:**
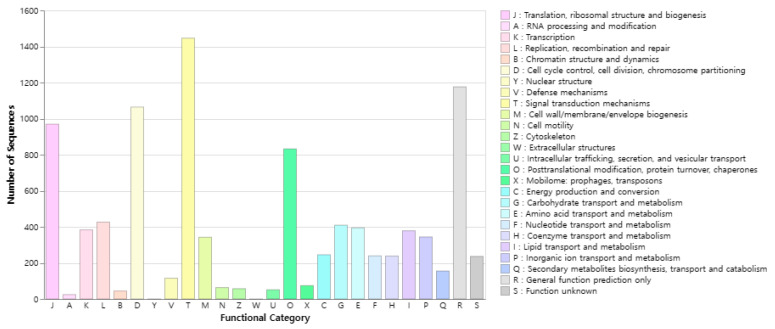
COG functional classification information in *R. uyekii*. Genes were assigned to 26 categories, excluding the “extracellular structure” category.

**Figure 4 ijms-25-06689-f004:**
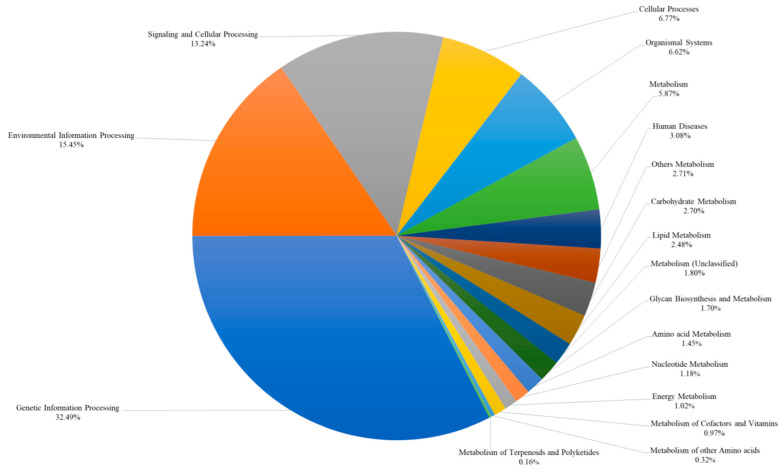
Percentage of KEGG ontology (KO) terms annotated in the *R. uyekii* genetic data set. Genes annotated in the Kyoto Encyclopedia of Genes and Genomes (KEGG) database are grouped into major functional categories based on the annotated pathways.

**Figure 5 ijms-25-06689-f005:**
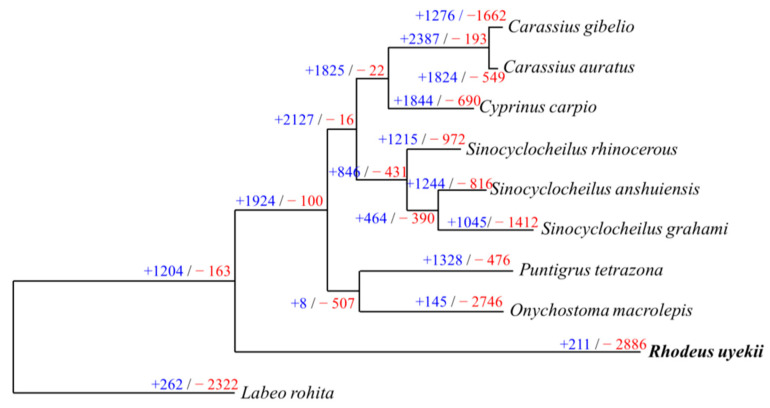
A phylogenetic tree of 10 species, including fish species. Using CAFE5, gene family expansions (+) and contractions (−) were calculated at each ancestral node and divergence and species.

**Figure 6 ijms-25-06689-f006:**
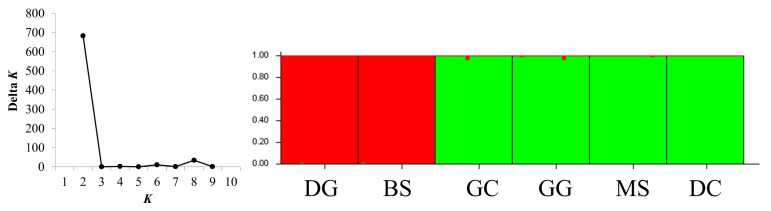
Population structure information through STRUCTURE analysis of *R. uyekii*.

**Figure 7 ijms-25-06689-f007:**
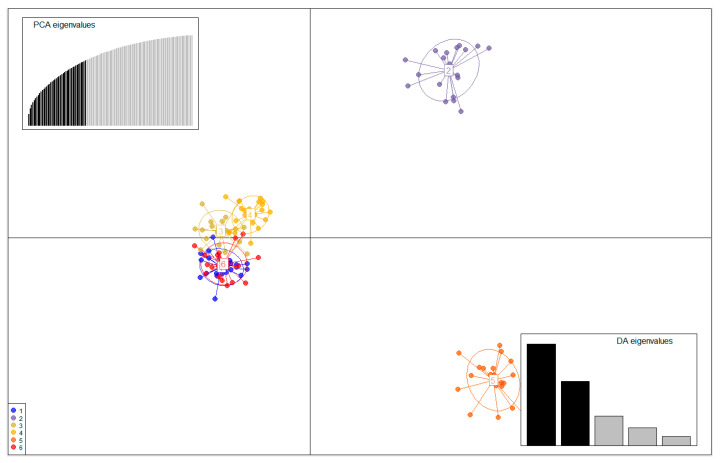
Discriminant analysis of principal components (DAPC) results that can analyze the structure of the population based on a nonmodel. 1: DC, 2: BS, 3: MS,4: GC, 5: DG, 6: GG.

**Figure 8 ijms-25-06689-f008:**
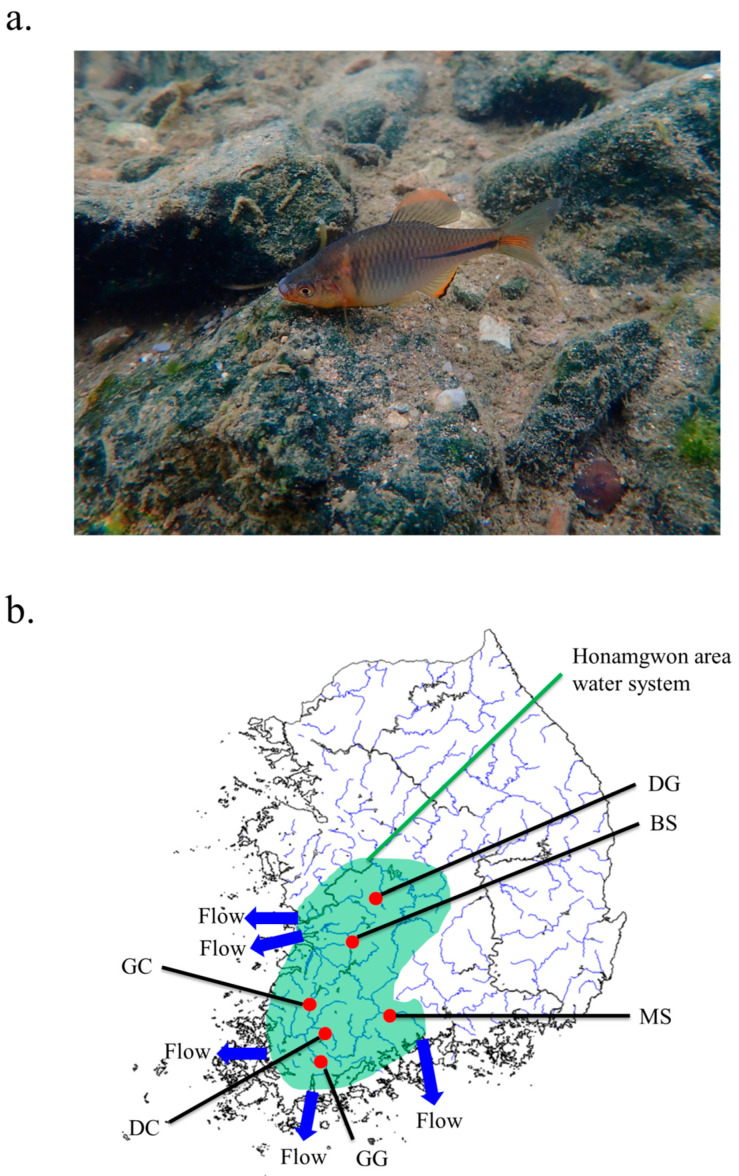
Photographs of specimens of *Rhodeus uyekii* used in the study (**a**). Location of sites for *R. uyekii* sample collection (**b**). The samples were collected from Dugyecheon (DG), Bongseocheon (BS), Gaecheon (GC), Geumgangcheon (GG), Masancheon (MS), Daechocheon (DC).

**Table 1 ijms-25-06689-t001:** Summary information of raw data for long and short read sequencing in *Rhodeus uyekii.*

Short Reads	Total Reads	Total Reads Length (bp)	Q20 (%)	Q30 (%)
Raw data	979,241,650	147,865,488,978	94.16	88.24
Filtered data	504,366,512	76,000,788,304	99.50	97.77
			N50 (bp)	Mean reads
HiFi reads data	5,573,756	43,361,492,973	8660	7779

**Table 2 ijms-25-06689-t002:** Summary information of genome assembly of *R. uyekii.*

Assembler	HGAP4
Contig	2652
Total contig bases (bp)	894,559,954
N50 (bp)	2,964,661
Max length (bp)	14,422,421
Average length (bp)	337,315

**Table 3 ijms-25-06689-t003:** Summary information of genome size of *R. uyekii.*

K-mer (21)	Minimum	Maximum
Heterozygosity (%)	0.900	0.909
Genome haploid length	635,909,967	636,773,012
Genome repeat length	76,383,271	76,486,937
Model fit (%)	95.52	97.02
Read error rate (%)	0.091	0.091

**Table 4 ijms-25-06689-t004:** Benchmarking Universal Single-Copy Orthologs analysis results of the two assembly methods.

Status	Parameter
Complete BUSCO (C)	3259 (97.2%)
Complete and single-copy BUSCO (S)	3220 (96.0%)
Complete and duplicated BUSCO (D)	39 (1.2%)
Fragmented BUSCO (F)	35 (1.0%)
Missing BUSCO (M)	60 (1.8%)
Total BUSCO groups searched	3354 (100%)

**Table 5 ijms-25-06689-t005:** Summary of protein gene predictions and annotations for *R. uyekii.*

	Numbers of Gene	Total Length (bp)
Predicted proteins	31,166	85,997,067
Database	Number	Percent (%)
EggNOG	29,896	95.93
InterPro	24,820	79.64
Pfam	23,354	74.93
COG	9704	31.14
KEGG	17,614	56.52

**Table 6 ijms-25-06689-t006:** Genetic diversity based on 21 microsatellite loci of *R. uyekii.*

ID	*n*	*N_A_*	*H* _O_	*H* _E_	*P* _HWE_	*F* _IS_
DG	20	7.6	0.496	0.587	0.000 ***	0.107 **
BS	20	8.0	0.568	0.695	0.000 ***	0.252 ***
GC	20	9.9	0.583	0.746	0.000 ***	0.234 ***
GG	20	8.9	0.642	0.783	0.000 ***	0.161 ***
MS	20	8.7	0.588	0.711	0.000 ***	0.144 ***
DC	20	8.2	0.612	0.737	0.000 ***	0.110 *

*n*: Number of samples, *N*_A_: Number of alleles, *H*_O_: Observed heterozygosity, *H*_E_: Expected heterozygosity, *P*_HWE_: Hardy-weinberg equilibrium *p*-value, * *p* < 0.05, ** *p* < 0.01, *** *p* < 0.001.

**Table 7 ijms-25-06689-t007:** Summary statistics regarding bottleneck signature for populations at microsatellites of *R. uyekii.*

Population ID	*n*	Wilcoxon Sign-Rank Test			
		P_IAM_	P_TPM_	P_SMM_	Mode-Shift
DG	20	0.004 **	0.367	0.822	L-shaped
BS	20	0.000 ***	0.018 *	0.108	L-shaped
GC	20	0.001 **	0.620	0.804	L-shaped
GG	20	0.000 ***	0.341	0.822	L-shaped
MS	20	0.000 ***	0.021 *	0.216	L-shaped
DC	20	0.000 ***	0.069	0.329	L-shaped

*n*: Numbers of Sample, P_IAM_: *p* value of bottleneck test using infinite allele mutation model, P_TPM_: *p* value of bottleneck test using two-phase mutation model (10% variance and 90% proportions of SMM), P_SMM_: *p* value of bottleneck test using stepwise mutation model, * *p* < 0.05, ** *p* < 0.01, *** *p* < 0.001.

**Table 8 ijms-25-06689-t008:** *F*_ST_ among populations according to microsatellite analysis of *R. uyekii.*

	DG	BS	GC	GG	MS	DC
DG	-	0.000	0.000	0.000	0.000	0.000
BS	0.177	-	0.000	0.000	0.000	0.000
GC	0.147	0.075	-	0.000	0.000	0.000
GG	0.193	0.116	0.070	-	0.000	0.000
MS	0.153	0.119	0.045	0.106	-	0.000
DC	0.162	0.111	0.041	0.096	0.070	-

## Data Availability

The assembled genomic sequence has been deposited in the GenBank WGS database, with accession number JAWRVZ000000000 (SAMN38042216). Twenty-one microsatellite markers were deposited in GenBank (OR753515-OR753535).

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
