# Peer review of "Genetic Diversity and Population Structure of Rhodeus uyekii in the Republic of Korea Revealed by Microsatellite Markers from Whole Genome Assembly"

_ijms, 2024, doi:10.3390/ijms25126689_

Round 1

Reviewer 1 Report

Comments and Suggestions for Authors

The study describes and enlightens for the first time the draft genome of an endemic fish from Korea providing also microsatellite markers that could be useful in future population structure studies as well as conservation efforts. Keeping in mind that there is a plethora of species, the genome of which is still unknown, it present a significant contribution to the literature. However, the way written and the title as well is confusing. The authors have to modify the introduction, discussion and the title in order to focus clearly in both directions. For instance, the title, as it is, refers only to microsatellite annotation, not mentioning anything concerning population genetics. Similarly the discussion of the population genetics part lacks demographic history of populations that could be related with genetic results. The manuscript has to be improved taking into consideration these issues

You can find detailed comments below

Line 18. macrolepis no Macrolepis, please correct

As written in the abstract it is like PCA to be based only on 10 samples. Please clarify that populations also were collected. Otherwise population genetic analyses would pointless

line 35: endemic where? In Korea? In a lake? Please be specific 

Lines 47-50: I suggest to add “(Figure 1)” at this point

line 75: were stored, not were using, please correct

Where are the primer sequences? In order for microstaellites to be used from other researchers in the future, primers have to be provided

Figure 2: Please indicate the position of the region in a larger map, to show the location. Not all readers are familiar 

The discussion is proportionally small and the conclusion is very large. Generally, the population genetics part has to be more extensively discussed. Are there any human mediated transportations of this fish among the sampling sites? Is there any other factor that enhances the clustering into three groups? Where is gene flow attributed? Human mediated translocations may alter genetic structure (Giantsis et al. https://doi.org/10.1017/S0025315414000174), could it be the case here? Please discuss this more extensively

Author Response

Response author

Q1. Line 18. macrolepis no Macrolepis, please correct

A1. Thanks for the review. The manuscript was revised to reflect the comments.

Q2. As written in the abstract it is like PCA to be based only on 10 samples. Please clarify that populations also were collected. Otherwise population genetic analyses would pointless

A2. Thanks for the review. Methods for population genetic analysis from samples have been added to the abstract.

[Population genetic analysis analyzed genetic diversity and genetic structure from 120 individuals from 6 populations.]

A3. line 35: endemic where? In Korea? In a lake? Please be specific

A3. Thanks for the review. The manuscript was revised to reflect your comments.

 [Korean rose bitterling R. uyekii is an endemic in Korea and common freshwater fish that in-habits rivers and tributaries such as Geumgang, Mangyeonggang, Yeongsangang, Seomjingang, and Tamjingang rivers in the Honamkwon region.]

Q4. Lines 47-50: I suggest to add “(Figure 1)” at this point

A4. Thanks for the review. Based on comments from other reviewers, Figures 1 and 2 have been merged.

Q5. line 75: were stored, not were using, please correct

A5. Thanks for the review. The manuscript was revised to reflect your comments.

 [The collected samples were placed in 99.9% ethanol and stored in a deep freezer (-80°C).]

Q6. Where are the primer sequences? In order for microstaellites to be used from other researchers in the future, primers have to be provided

A6. Thanks for the review. Primer information is in the supplementary material.

 [Table S5. Characteristic and diversity information for the 21 microsatellite loci of R. uyekii]

Q7. Figure 2: Please indicate the position of the region in a larger map, to show the location. Not all readers are familiar

A7. Thanks for the review. This was replaced by changing the map to draining water at each point.

Q8. The discussion is proportionally small and the conclusion is very large. Generally, the population genetics part has to be more extensively discussed. Are there any human mediated transportations of this fish among the sampling sites? Is there any other factor that enhances the clustering into three groups? Where is gene flow attributed? Human mediated translocations may alter genetic structure (Giantsis et al. https://doi.org/10.1017/S0025315414000174), could it be the case here? Please discuss this more extensively

A8. Thanks for the review. We briefly described some of the conclusions and reflected the population genetics section in the discussion section.

[Coreoleuciscus splendidus, similar to R. uyekii, inhabits most water bodies in South Korea. Coreoleuciscus splendidus was reported to be divided into two populations in the Honamkwon region basin [7]. In this study, the R. uyekii population showed significant genetic differences, and the genetic structure was divided into two populations (DG, BS vs GC, GG, MS, DC) in the STRUCTURE results. However, DAPC results showed that the population was divided into three (BG vs. DG vs. GC, GG, MS, DC) on a non-model basis. Freshwater fish of Korean Peninsula typically exhibit genetic variation across geographic watersheds, with little genetic variation within the same watershed [7]. The FST results showed significantly higher genetic differentiation in DG vs BS, FST=0.177, suggesting that the six populations are genetically three populations. This means that genetic differentiation exists between Geumgang and Mangyeonggang rivers, which is strongly supported in Pseudopungtungia nigra [4].]

Reviewer 2 Report

Comments and Suggestions for Authors

The manuscript reported a genome sequencing of the Rhodus uyekii and study genetic structure of wild population by microsatellite markers obtained from next-generation sequencing. It will be interesting for germplasm conservation and evolutionary researches for Rhodus and related fish species. The description of main text and the figures were fine. It is suitable for publication considering minor revisions.

1.      The title may suggest to be revised such as: Genetic Diversity and Population Structure of Rhodeus uyekii in Korea Revealed by Microsatellite Markers from Whole Genome Assembly.

2.      Abstract:Lane 25, the description of BS vs DC vs GG, GC, MS, DC is not clear.

3.      Results: Figures 1 and 2 may be merged as one figure. Table 3 and Table 4 may put in supplementary tables.

4.      Lane 338: "Principal component discriminant analysis" may be corrected as "Discriminant analysis of principal components".

5.      The reference list will be edited following the instruction of the Journal.

Author Response

Response author

Q1.      The title may suggest to be revised such as: Genetic Diversity and Population Structure of Rhodeus uyekii in Korea Revealed by Microsatellite Markers from Whole Genome Assembly.

A1. Thanks for the review. The manuscript was revised to reflect your comments.

Q2.      Abstract:Lane 25, the description of BS vs DC vs GG, GC, MS, DC is not clear.

A2. Thanks for the review. The manuscript was revised to reflect your comments.

[Discriminant analysis of principal components According to the analysis method, the population was divided into three populations (BS vs. DC vs. GG, GC, MS, DC).]

Q3.      Results: Figures 1 and 2 may be merged as one figure. Table 3 and Table 4 may put in supplementary tables.

A3. Thanks for the review. Figures 1 and 2 have been incorporated, and Tables 3 and 4 have not been incorporated as supplementary material because they represent the genome size and quality of the assembled data. Please understand.

Q4.      Lane 338: "Principal component discriminant analysis" may be corrected as "Discriminant analysis of principal components".

A4. Thanks for the review. The manuscript was revised to reflect your comments.

Q5.      The reference list will be edited following the instruction of the Journal.

A5. Thanks for the review. The manuscript was revised to reflect your comments.

Round 2

Reviewer 1 Report

Comments and Suggestions for Authors

I am satisfied with the modifications performed in the study

Author Response

Q1. I am satisfied with the modifications performed in the study

A1. Thanks for the review.